# Co-Carbonization of Discard Coal with Waste Polyethylene Terephthalate towards the Preparation of Metallurgical Coke

**DOI:** 10.3390/ma16072782

**Published:** 2023-03-30

**Authors:** Sonwabo E. Bambalaza, Buhle S. Xakalashe, Yolindi Coetsee, Pieter G. van Zyl, Xoliswa L. Dyosiba, Nicholas M. Musyoka, Joalet D. Steenkamp

**Affiliations:** 1Pyrometallurgy Division, Mintek, 200 Malibongwe Drive, Praegville, Johannesburg 2125, South Africa; 2Chemical Resource Beneficiation, North West University, Potchefstroom 2531, South Africa; 3HySA Infrastructure Centre of Competence, Centre for Nanostructures and Advanced Materials (CeNAM), Chemicals Cluster, Council for Scientific and Industrial Research (CSIR), Pretoria 0001, South Africa; 4Expert Process Solutions—Glencore, Pyrometallurgy and Furnace Integrity, 6 Edison Road, Falconbridge, ON P0M 1S0, Canada

**Keywords:** discard coal, waste polyethylene terephthalate, co-carbonization, waste utilization, metallurgical coke

## Abstract

Waste plastics such as polyethylene terephthalate (w-PET) and stockpiled discard coal (d-coal) pose a global environmental threat as they are disposed of in large quantities as solid waste into landfills and are particularly hazardous due to spontaneous combustion of d-coal that produces greenhouse gases (GHG) and the non-biodegradability of w-PET plastic products. This study reports on the development of a composite material, prepared from w-PET and d-coal, with physical and chemical properties similar to that of metallurgical coke. The w-PET/d-coal composite was synthesized via a co-carbonization process at 700 °C under a constant flow of nitrogen gas. Proximate analysis results showed that a carbonized w-PET/d-coal composite could attain up to 35% improvement in fixed carbon content compared to its d-coal counterpart, such that an initial fixed carbon content of 14–75% in carbonized discard coal could be improved to 49–86% in carbonized w-PET/d-coal composites. The results clearly demonstrate the role of d-coal ash on the degree of thermo-catalytic conversion of w-PET to solid carbon, showing that the yield of carbon derived from w-PET (i.e., c-PET) was proportional to the ash content of d-coal. Furthermore, the chemical and physical characterization of the composition and structure of the c-PET/d-coal composite showed evidence of mainly graphitized carbon and a post-carbonization caking ability similar to that of metallurgical coke. The results obtained in this study show potential for the use of waste raw materials, w-PET and d-coal, towards the development of an eco-friendly reductant with comparable chemical and physical properties to metallurgical coke.

## 1. Introduction

As the world moves towards clean and renewable energy technologies, such as solar and wind, the demand for steel production also increases with these markets. The structural components of wind turbines are made up of 90% steel, and the production of high purity silicon metal (solar grade Si) both require carbothermic reduction smelting processes during their extraction [1,2,3]. This has a direct impact on the demand for metallurgical coke as it is the reductant of choice in both steelmaking and silicon smelting. The shift towards clean renewable power generation technologies is inevitably based on the adverse environmental effects caused by increased carbon dioxide (CO_2_) atmospheric levels [4,5]. At a local landscape, the depletion of South African hard coking coals places the country at a disadvantage as a role player in the metallurgical coke industry. This is evidenced in the high rate of high grade coke imports from countries such as Australia. In a typical steel manufacturing process, coking coal accounts for about 27% of total costs [6,7], and since 2008, the South African steelmaking companies consumed about 4 Mt of coking coal, of which half came from Australian imports [8]. The shortage in supply was attributed to the dependence of the local steelmaking industry on the major producers of coking coal in South Africa producing 2 Mt per annum [9].

The precursor material(s) of choice for industrial coke-making are typically hard coking coals obtained from coal mining operations. This, therefore, makes the overall coke-making process contribute CO_2_ emissions through a combination of coal mining and subsequent pyrolysis of hard coking coals to produce metallurgical coke as the final product [10,11]. In a study by Miller et al. (2017) [12], the magnitude of CO_2_ emissions resulting from both underground and surface mining of hard coking coals showed emissions of 170–430 kg CO_2_ and 5–35 kg CO_2_ mass equivalent per tonne of coal mined, respectively. In addition, non-beneficiated coal that contains high ash content (termed discard coal) finds little economic value, poses a danger for human health by polluting the soil, and presents a long-term risk of spontaneous combustion when stockpiled in open storage pits [13]. Due to the high ash content and composition of d-coal, an opportunity may arise to utilize discard coals by possibly exploiting some of the principles from thermo-catalytic synthesis of carbon using hydrocarbon precursors in the presence of inorganic catalysts [14,15,16,17,18]. Indeed, the literature shows that carbon has been synthesized from plastic waste and also the co-pyrolysis of coals with plastic waste has been demonstrated [19,20,21]. A combination of some aspects from the two methods can be exploited to possibly use d-coals and plastic waste as feedstock in a coke-making process, forming an eco-friendly pathway towards preparation of metallurgical coke.

According to the “National inventory discard and duff coal” report of 2001 [8] and related research studies [9,22], South African d-coal is a major environmental problem that constitutes about 1500 Mt in coal mine dumps and landfills. The production of d-coal is reported at an annual rate greater than 42 million tonnes with about 94% of the discard dumps constituted of bituminous d-coals and the remaining 6% made up of anthracitic d-coals. South African d-coals typically contain ash contents ranging from 40 to 50%, volatile matter between 16 and 20%, and fixed carbon content between 10 and 42%. The composition of the ash may typically consist of varying amounts of acidic and basic metal oxides such as quartz (SiO_2_), alumina (Al_2_O_3_), iron oxide (Fe_2_O_3_), magnesium oxide (MgO), and calcium oxide (CaO) [8,23,24].

Metal oxides have been shown to be useful catalysts in the synthesis of carbonaceous materials from hydrocarbon precursors [18,25]. The catalytic activity of metal oxides occurs under varying conditions of temperature, pressure, and the type of hydrocarbon reactants. In general, acidic oxides contain catalytic sites which can catalyze the protonation of unsaturated hydrocarbons, doubly bonded alkenes, to form secondary cations followed by double bond or skeletal isomerization. In saturated hydrocarbons, alkanes are protonated into penta-coordinated carbocations, aromatic hydrocarbons are protonated into benzenium cations, and alcohols to oxonium ions. Other acidic catalytic sites, on the other hand, favor ring opening reactions of cyclic hydrocarbons and the formation of epoxides in oxygen-containing hydrocarbons. The Brønsted base catalytic sites are able to remove hydrogen ions (protons) from alkenes to form allylic carbanions, resulting in the formation of double bonds or skeletal isomerization. Both acid and base sites can also act cooperatively such as in alcohol dehydration reactions or aldol condensation [26,27]. These different types of reactions of hydrocarbons onto metal oxide catalysts form part of the catalytic cracking of large polymeric hydrocarbons into smaller oligomeric compounds and deposition of solid carbon as demonstrated by Park et al. [16].

The current study, thus, focuses on the development of a w-PET and d-coal composite material that potentially shows a fixed carbon content, ash content, and mechanical strength similar to metallurgical coke as an eco-friendly alternative to metallurgical coke. The desired w-PET/d-coal composite could be attainable by exploiting thermal reactions between a typical hydrocarbon precursor, such as waste plastic and metal oxides (present in d-coal ash) to produce a carbonaceous solid residue. The selection of a suitable d-coal originating from coking, bituminous, or anthracitic coals could have the potential to develop a final high grade w-PET/d-coal product with metallurgical coke properties suitable for application as a reductant in submerged- and open-arc ferroalloy production processes.

In support of the aforementioned concept, the literature reports on the use of household waste plastic in an industrial coke making process. Some of the household waste plastics used are, polyethylene (PE), polystyrene (PS), polypropylene (PP), and polyethylene terephthalate (PET) [21,28]. The use of plastics in coke-making originates from the high-temperature thermal behavior (melting, decomposition, and production of char) of plastics under conditions with a constant flow of an inert gas, e.g., nitrogen or argon. Under these conditions, plastics such as PET and PS are converted into a carbonaceous char residue at temperatures above 400 °C [29,30]. The systematic review by Nomura et al. [21] showed that the combination of a small amount of plastic material (ca. 1–3% by mass) with coking coals can be successfully incorporated into conventional coke ovens to produce high quality metallurgical coke without compromising the coke properties such as hardness and coke fluidity.

In addition, the use of coal ash for the synthesis of different types of carbons has been reported for the preparation of carbon nanomaterials such as carbon nanofibers (CNFs) and carbon nanotubes (CNTs) [31,32]. Rambau et al. [31] reported on the use of coal fly ash (CFA) as a catalyst for the synthesis of CNFs/CNTs from a waste tyre pyrolysis oil carbon precursor. The study used a horizontal tube furnace, generally referred to as the carbon vapor deposition (CVD) method, which introduces a precursor hydrocarbon gas or pyrolysis oil aerosols into a pre-heated CFA catalyst, ca. 650–700 °C, under a constant flow of nitrogen gas. At the designated temperature, the gas/pyrolysis oil is catalytically converted into solid carbon and volatiles by reacting with the metal oxides present in the CFA. Other related studies [33] have reported on the presence of specific metal oxides such as the aluminosilicate mullite, further improving the carbon yield of CVD-grown carbon materials. The CVD method has been well-documented and successfully applied in the preparation of carbon nanomaterials for various applications such as gas storage, waste water treatment, and in supercapacitors [31,34,35].

In this study, both concepts of catalytic carbonization using coal ash (from primary and secondary sources) and the conversion of hydrocarbons into carbon/char will be exploited through the combination of w-PET and high-ash-content d-coals. The amount of w-PET in the composite is envisaged at loadings in excess to the amount of d-coal so as to potentially maximize the degree of carbon deposition possible from the carbonization of w-PET/d-coal composites at elevated temperatures.

## 2. Materials and Methods

### 2.1. Materials

Recycled polyethylene terephthalate pellets (PETCO, Cape Town, South Africa), run of mine (ROM) blend product (Exxaro Resources, Lephalale, South Africa), discard spiral middlings (Exxaro Resources, South Africa), discard belt filter cake (Exxaro Resources, South Africa), and hydrochloric acid (32% HCl, *v*/*v*,) and hydrofluoric acid (40% HF, *v*/*v*), both supplied by Associated Chemical Enterprises (Johannesburg, South Africa).

### 2.2. Methods

The carbonization procedures used in this study were adapted from the work of Rambau et al. [31] with minor adjustments specific to this study. The procedures are divided into three categories, namely: (i) direct carbonization; (ii) co-carbonization with d-coal ash; and (iii) co-carbonization with three different d-coal samples. The difference between direct carbonization and co-carbonization is the former involves the carbonization of only w-PET, whereas in the latter, there is a combination of w-PET with d-coal ash or d-coal, where the d-coal ash is obtained from combustion of the d-coal samples. Prior to the carbonization processes, sample preparation was carried out to obtain composites of w-PET/d-coal ash and w-PET/d-coal.

#### 2.2.1. Sample Preparation of w-PET/d-Coal Ash and w-PET/d-Coal Composites Prior to Carbonization

In the preparation of d-coal ash, combustion of d-coal was carried out in a muffle furnace (Lenton BRF 15/5 model) at 950 °C and held at the target temperature for 2 h. After cooling, the d-coal ash was then used to prepare a series of w-PET/d-coal ash mixtures containing w-PET to d-coal ash mass ratios of 1:1; 1:2; 3:2; 3:1; and 4:1 in a total sample weight of 20 g. A w-PET/d-coal ash monolith was then prepared by melting each w-PET/d-coal ash mixture at 350 °C for 10–15 min in a cylindrical crucible, followed by stirring the molten w-PET/d-coal ash mixture with an iron rod before transferring to a crucible boat, as shown in Figure 1. During stirring, it was important to minimize solidification and ensure to keep sample losses at a minimum (<10 wt%). The monolithic form of w-PET/d-coal ash ensures that d-coal ash powder particles are embedded on the w-PET surface prior to co-carbonization.

To prepare w-PET/d-coal composites, there were three (3) d-coal types used in the study, namely a run of mine (ROM) coal blend, a belt filter cake (BFC), and spiral middlings (SM). Firstly, each coal type was used on a dry basis by removing all moisture through drying in a conventional oven at 100 °C for 72 h. The dried d-coal powder was then milled in a Dickie & Stockler TS-250 mill (Dickie & Stockler (Pty) Ltd., Johannesburg, South Africa) down to an appropriate size range conducive for mixing with w-PET, as shown in Figure 2.

The milled d-coal, with particle size range of +0.25 mm to −1.00 mm, was mixed with w-PET pellets in an alumina crucible cup and melted to form a w-PET/d-coal monolith according to the above-mentioned procedure.

#### 2.2.2. Carbonization Process: Direct Carbonization and Co-carbonization

A generalized carbonization process is shown in Figure 3 highlighting the major steps during the direct- and co-carbonization methods employed on w-PET, w-PET/d-coal ash, and w-PET/d-coal samples. In a typical procedure, a 20 g sample was placed in a ceramic boat and heated from room temperature, under 0.5 L/min nitrogen flow, at an optimum heating rate of 5 °C/min and up to a final temperature of 700 °C, holding at the final temperature for 15 min (zone A and B). After the carbonization, the sample was subjected to rapid cooling by removing it at 700 °C and placing it onto a steel tray at ambient conditions (zone C). The compositions of the carbonized mixture/composite materials are given in Appendix A.

After completion of the carbonization, the yield of carbon derived from w-PET was determined. In direct carbonization, the final product was weighed using an analytical balance to the nearest 0.1 decimal and the residue yield of carbonized PET (c-PET) calculated using Equation (1):(1)$$R=\frac{m_{f}}{m_{i}}\times 100\%$$

*R*—residue yield (%)*m_i_*—initial mass of w-PET before carbonization (g)*m_f_*—final mass of product (c-PET) after carbonization (g)

For co-carbonization of w-PET/d-coal composites, the carbon yield of each sample was determined by thermogravimetric analysis (TGA), which is detailed in Section 2.3. In addition, for the purposes of obtaining free-standing c-PET particles, a demineralization step was conducted on the sample giving the highest total c-PET content (based on TGA results). In this step, a two-step acid washing method was employed with the first step involving stirring the c-PET/d-coal ash product in concentrated HCl (32% *v*/*v*) for 24 h at room temperature. This was performed to remove metal oxides other than SiO_2_ from the d-coal ash. The undissolved material was then recovered by vacuum filtration and washed with distilled water until the resultant filtrate solution reached pH 7. To remove the remaining SiO_2_, the product was then washed in HF (40% *v*/*v*) for 24 h at room temperature. After another washing with 3 L of distilled water, the collected c-PET was then oven-dried under vacuum at 90 °C for 24 h. The de-mineralization step was performed for the purposes of microstructural and morphological analysis of the c-PET product.

### 2.3. Characterization

The chemical composition of each of the d-coal ashes was determined by powder X-ray diffraction (PXRD) analysis. In a typical procedure, the d-coal ash was ground to a fine powder using an agate mortar and pestle prior to loading onto a glass sample holder. The sample was then placed in a Rigaku Ultima IV X-ray diffractometer with 0.154 nm Ni-filtered Cu K_α_ radiation (40 kV and 30 mA). Each sample was scanned from 2θ angles of 3–90° at a scan rate of 2°/min. To quantify the crystalline phases obtained in each d-coal ash sample, a quantitative XRD (QXRD) was used for analysis of the ash composition. The samples were placed in a Bruker D8 diffractometer equipped with a Lynxeye detector and Fe-filtered Co K_α_ radiation. The sample was run with a step size of 0.02 degrees 2θ with a counting time of 3 s per step from 3 to 80° 2θ. Phases were identified using Bruker EVA software and further quantified with the TOPAS software package. The thermal decomposition of the raw materials was compared to that of the product using Linseis STA PT1600 Thermogravimetric Analyzer (TG) coupled to a Differential Scanning Calorimeter (DSC) (Linseis Messgeräte GmbH, Selb, Germany). Prior to the TGA, each sample of d-coal and c-PET/d-coal ash or c-PET/d-coal was milled for 1 min using a Dickie & Stockler TS-250 mill with a pneumatic clamp. A ~20 mg powder sample was then placed in an alumina crucible and heated at a ramp rate of 10 °C/min to 1000 °C under static air or pyrolysis under argon flow of 3 L/h. The TGA data was used to determine the thermal behavior of each d-coal and c-PET/d-coal ash or c-PET/d-coal sample and calculate the fixed carbon content in each sample before and after carbonization. Raman spectroscopy, on a laser confocal Raman micro-spectrometer (JY LabRam HR 800), was used to investigate the electronic structure of carbon atoms giving information about the degree of defects and extent of multilayered graphitic groups in the carbon product. Raman analysis was only performed on de-mineralized c-PET powder as it represents free-standing carbon particles derived directly from the carbonization of w-PET, using a Raman excitation wavelength of 532 nm and scanned from 100–3000 cm^−1^. The microstructure of de-mineralized c-PET was analyzed using scanning electron microscopy (SEM) and to study the 3-dimensional morphologies, the SEM imaging was performed using an Auriga cobra focused-ion beam scanning electron microscope (FIB-SEM), with each sample being mounted onto carbon tape and coated with a layer of chromium (Cr) to prevent possible charging prior to each analysis. The cross-sectional surface and qualitative chemical composition of de-mineralized c-PET particles was performed on a Zeiss EVO MA 15 SEM, coupled to a Bruker Energy Dispersive X-ray Spectrometer (EDX). Each c-PET sample was coated in aluminum (Al) to ensure good carbon (C) peak intensities on the EDX spectrum unhindered by C coating. It should be noted, however, that the Al coating may slightly affect analyses of phases in which Al is present, but since only an indication of the impurity compositions was required, this was deemed suitable. The mechanical performance of the c-PET/d-coal composite was compared to that of a commercial metallurgical coke sample via a compressive strength test method. The test was performed using an Instron tensometer 3366 model equipped with Bluehill 3 software. Using the ISO 4700:2007 compressive strength test method, the typical testing procedure involved placing a lump of coke or c-PET/d-coal and compressing it under a load moving at a cross-head speed of 10 mm/min. The maximum load recorded at failure was recorded for each sample. In this particular case, the compressive strength was not calculated due to the highly irregular shape of the metallurgical coke lumpy particles, such that the diameter of the sample could not be accurately measured by conventional methods. For the purposes of this study and comparisons, only the maximum load at failure (measured in Newtons) was recorded and used for compressive strength comparisons.

In the results Section 3.1 and Section 3.2, the study focuses on the characterization of c-PET products derived from direct carbonization of w-PET in comparison to co-carbonization of w-PET/d-coal ash and w-PET/d-coal composites. Their comparisons are based on the thermal behavior, chemical structure (composition, electronic and crystalline structure), particle morphologies/distribution, and the carbon yields of the different c-PET products.

## 3. Results and Discussion

### 3.1. Direct Carbonization of w-PET in Comparison to Co-carbonization of w-PET/d-Coal Ash Composites

The direct carbonization of w-PET is well documented in the literature [36,37,38] and is typically characterized by the formation of a solid char/carbon residue (ca. 10 wt% yield) in the 400–500 °C temperature range upon its thermal treatment at elevated temperatures under constant flow of an inert gas. In Figure 4a, w-PET samples were heated in a TG from ambient temperature up to 1000 °C under different gas flows, argon flow to represent inert conditions, and static air representing oxidative conditions. The thermal behavior obtained under argon flow showed that a peak decomposition was observable at 450 °C and the sample produced a final char (denoted as c-PET) residue yield of 12 wt%. In the presence of air; however, the c-PET residue decomposed at 700 °C due to combustion (Figure 4a and Appendix A).

The observed results are consistent with the findings of Ko et al. [36] related to the direct carbonization of w-PET; however, in addition to direct carbonization, co-carbonization of w-PET with d-coal ash was performed by heat treatment of a w-PET/d-coal ash mixture under nitrogen flow as stipulated in Section 2.2.2.

In Figure 4b, it can be seen that co-carbonization of w-PET with varying amounts of d-coal ash produced differing c-PET yields. From the thermogravimetric curves, it can be seen that the d-coal ash undergoes negligible thermal decomposition when heated up to 1000 °C, but the c-PET/d-ash composites undergo major decomposition between 450 and 700 °C. These results are expected as the bulk composition of the d-coal ash was identified to be made up of Fe_2_O_3_ in the form of hematite, mullite (3Al_2_O_3_·2SiO_2_), SiO_2_ as quartz (Appendix A), and other minerals which have melting and decomposition temperatures greater than 1000 °C. On the other hand, carbon typically decomposes at 500–800 °C as it reacts with oxygen in air to form CO_2_ during combustion. This forms the basis for estimating the amount of carbon formed in each c-PET/d-ash mixture using TGA data. The residue yield obtained for each c-PET/d-ash composite indicates the amount of carbonaceous material burnt off during the decomposition of the c-PET/ash composite and as a result gives an estimate of the amount of carbon deposited from the carbonization of w-PET in each mixture. The c-PET yield for each sample was, therefore, calculated using Equations (2) and (3) by measuring the residue yield of each sample as the weight loss at 950 °C and calculating the amount of volatiles as the total weight loss in each sample:(2)V=100wt%−R

*V*—Total weight loss (volatiles yield) at 950 °C (wt%)*R*—Residue yield at 950 °C (wt%)


(3)
$${\left[C\right]}_{PET}=V_{PET}-V_{ASH}$$


[*C*]*_PET_*—Fixed carbon yield derived from w-PET after co-carbonization (wt%)*V_PET_*—Total weight loss (volatiles yield) of c-PET/d-coal ash composite at 950 °C (wt%)*V_ASH_*—Total weight loss (volatiles yield) of d-coal ash sample at 950 °C (wt%)

In Table 1, the calculated c-PET yields for the w-PET/d-coal ash composites were all found to be greater than 15 wt%, with the highest c-PET yield of 46.2 wt% calculated for the 3:1 w-PET/d-coal ash composite. This means the co-carbonization of w-PET/d-coal ash composites could potentially produce a carbon yield of up to 3.8 times, by weight, compared to the amount of carbon produced from direct carbonization of w-PET under the same conditions specified in this study. The lower c-PET yield of 38.2 wt% obtained for the 4:1 w-PET/d-coal ash composite shows that a maximum c-PET yield was obtainable from co-carbonization of w-PET/d-coal ash composites, such that a w-PET/d-coal ash composite containing more than 75 wt% of w-PET did not result in an increased c-PET yield. Due to the high yield of c-PET obtained from the 3:1 w-PET/d-coal ash composite, it was this c-PET/d-coal ash composite that was de-mineralized and prepared for chemical characterization of free-standing c-PET particles and compared to those obtained via direct carbonization of w-PET.

#### 3.1.1. Chemical Structure Analysis of c-PET

The crystal structure of graphitic carbons (e.g., in graphene and graphite) typically gives rise to distinct and characteristic diffraction patterns that can be easily analyzed using the PXRD method. In this study, PXRD was specifically used as a qualitative tool to determine the presence/absence of graphitic carbons and their crystallinity in c-PET products prepared by direct carbonization of w-PET pellets in comparison to co-carbonization w-PET/d-coal ash composites.

A typical XRD pattern of graphitic carbons consists of diffraction peaks at 24–26°, 44°, and 81°, which represent the (0 0 2), (1 0 0), and (1 1 0) planes, respectively [39,40,41]. Analysis of these specific peaks gives information about the structure of graphite microcrystallites, including the average stack height or ordering of graphite planes, and the size of hexagonal ring structures. In general, the peak width of the (0 0 2) plane at 24–26° relates to a high microcrystallite ordering when the peak is narrow, whereas a broad peak typically relates to a less ordered microcrystallite structure. In addition, the higher and narrower the (1 0 0) and (1 1 0) planes are related to the large-sized hexagonal rings [39].

In Figure 5a, the crystal phase composition of c-PET, derived from direct carbonization of w-PET, shows two broad peaks at 2θ ~20–30° (peak at 25°) and 2θ ~40–50° (peak at 43°), which is typical of a low degree of crystalline phases in the carbon and under extremely broad peaks represent amorphous carbon phases [40,42]. The appearance of broad peaks in the c-PET sample is consistent with previous studies [36,40,42] which have shown that there is a very small degree of graphitization achievable from the direct carbonization of PET at temperatures below 1000 °C. The diffraction patterns in Figure 5b obtained for the demineralized c-PET/d-coal ash product have the same peak positions at 2θ ~25° and 43° as obtained for c-PET from direct carbonization; however, the peak widths of the demineralized c-PET/d-coal ash sample are narrower in comparison to the broader peaks obtained for c-PET, an indication of a higher degree of structural order or crystallinity in the graphite planes obtained from the co-carbonization of w-PET and d-coal ash. There are additional broad peaks evident at lower 2-theta angles, specifically at 2θ ~5–10° and 2θ ~10–20°, which can be attributed to a type of amorphous carbon allotrope present in the sample.

It is also useful to use Raman spectroscopy to gain information about the electronic structure of graphitic carbons.

In Figure 6a,b, the Raman spectroscopic analysis was used to determine the type of carbon that is formed from each of the carbonization processes. It is generally accepted that the Raman spectrum of graphitic carbons gives information about the degree of graphitization in the sample using three (3) specific peak positions at ca. 2690 cm^−1^, 1580 cm^−1^, and 1350 cm^−1^, known as the “2D band”, “G band”, and “D band”, respectively [43,44,45]. The intensity ratios, I_2D_/I_G_ and I_D_/I_G_, give structural information about the type of graphitized carbon. In samples with I_2D_/I_G_ < 1, there is a low degree of single graphene layers, whereas I_D_/I_G_ > 1 represents a large degree of structural disorder in the sample (e.g., strain or presence of functional groups on the carbon atoms). In this study, it is evident that both c-PET products from direct carbonization and co-carbonization of w-PET/d-coal ash composites did not produce a 2D band, thus showing the graphitized c-PET products as entirely made up of multilayered graphene layers (graphite). There is, however, a difference in the degree of structural disorder of these c-PET products, as it can be seen in Figure 6a,b and Appendix A that the I_D_/I_G_ ratio equals 1.40 for c-PET obtained by direct carbonization and only a I_D_/I_G_ ratio of 1.04 was obtained when w-PET was co-carbonized with d-coal ash. The lower I_D_/I_G_ ratio indicates the formation of more ordered carbon atoms when w-PET is co-carbonized with d-coal ash in comparison to its direct carbonization. The observed results from the Raman analysis are in agreement with the structural order differences deduced from the PXRD analysis of the samples (Figure 4). Both Raman and PXRD analyses provide strong evidence of differences in the chemical structures of c-PET products obtained via direct carbonization and co-carbonization methods, which may be substantiated through imaging the morphology of free standing c-PET particles using electron microscopy.

#### 3.1.2. Morphological Analysis of c-PET Particles

The imaging of c-PET particles was performed using FIB-SEM and elemental analysis obtained via SEM-EDX of the cross-sectional surface area of each c-PET product.

In Figure 7a, it is evident that the c-PET particles derived from the direct carbonization of w-PET are mainly irregularly shaped with no well-defined edges. This observation is also mostly evident in the cross-sectional image of the particles in Figure 7b. From the EDX mapping in Figure 7c,d, it can be seen that the bulk composition of the c-PET particles was made up of elemental carbon and no evidence of elemental oxygen atoms. It was, however, observed that there were small white particles distributed between the carbon particles. Using EDX spot analysis, it was found that the composition of these particles was found to consist of silica, aluminosilicate, calcium aluminosilicate, and titanium aluminosilicate (Appendix A), which could have been introduced as contaminants during the carbonization of w-PET. It is important to note that the contaminants or impurities are not present within the carbon particles as such an observation would imply the incorporation of a metal oxide catalyst during the direct carbonization of the c-PET product. The source of these impurities was thus due to residual inorganic solid particles on the crucible boat or could have been on the sample holder during sample preparation.

The imaging of de-mineralized c-PET/d-coal ash composite particles was investigated similarly to the aforementioned c-PET particles from direct carbonization of w-PET.

In Figure 8, the de-mineralized c-PET/d-coal ash product was found to consist of two main particle morphologies, spherical and irregularly shaped particles, as shown in Figure 8a,c (also see Appendix A), both having a bulk composition of elemental carbon. The detection of carbon spheres, a type of amorphous allotrope of carbon, shows clear evidence for the amorphous phase (2θ ~10–20°) observed in the PXRD pattern in Figure 5b. The irregularly shaped c-PET particles are characterized by sharp/defined edges and a layered appearance, suggesting a high degree of crystallinity/order within the microstructure of the particles.

The morphology of the cross-sectional surface, in Figure 8b,d, shows evidence of impurities within the carbon spheres and irregularly shaped particles. The EDX mapping in each sample show that the main impurity in carbon spheres was vanadium (V) and iron (Fe). The source of Fe can be attributed to the presence of hematite in the d-coal ash which could have remained in trace amounts following the acid-washing steps of the c-PET/d-coal ash product. Vanadium, on the other hand, does not seem to have a direct source as no vanadium mineral was detected in the d-coal ash by QXRD analysis (Appendix A). The vanadium could have originated from an impurity in the d-coal ash possibly below the detection limit of the QXRD which varies between 1 and 3%.

The experimental results obtained for the co-carbonization of w-PET/d-coal ash composites showed evidence for the formation of elemental carbon derived from w-PET under the process conditions specified in this study. The results show potential for co-carbonization of w-PET/d-coal composites where d-coal replaces d-coal ash as the substrate for carbon deposition. Prior to investigating carbon deposition on w-PET/d-coal composites, it was important to first determine the fixed carbon content of the raw d-coal samples such that the c-PET yield would be calculated from the difference between the fixed carbon content of the raw d-coal and that of the carbonized c-PET/d-coal composites (as per Equation (S2)). This was achieved by performing proximate analysis on carbonized samples of d-coal and c-PET/d-coal composites using traditional ASTM standard testing methods employed in coal petrography, and TGA thermograms to substantiate the ASTM experimental results by graphical representation. The proximate analysis of raw d-coal samples is given in Appendix A.

### 3.2. Co-Carbonization of w-PET/d-Coal Composites: Proximate Analysis

The proximate analysis results are presented based on TGA experiments (Figure 9) and ASTM standard methods (Appendix A). The TGA experiments were designed to closely resemble the method specifications as specified in ASTM standard method in terms of the heating rate, holding time at the target temperature, and gas flowrates (where applicable or necessary).

In Figure 9a, the thermal decomposition of each sample was analyzed under a constant heating rate of 10 °C/min and 3 L/min flow of argon, up to a final temperature of 1000 °C. It can be seen that for each sample there was a sharp weight loss from room temperature up to 100 °C, which is typically as a result of adsorbed water evaporating from the surface of the sample. The amount of surface water was found to be highest in the c-ROM blend sample, while the lowest weight loss was in c-PET/c-ROM blend. The results also show that the d-coal samples (c-ROM, c-BCF, and c-SM) contain higher surface water content in comparison to their composite c-PET/d-coal counterparts. This is because the raw d-coals were provided as wet samples, which were dried prior to carbonization. The onset decomposition temperature, described as the temperature at which 5% weight loss is observed [46], was found to be around 450–500 °C for c-ROM blend and c-BCF samples but greater than 650 °C for their c-PET/d-coal counterparts. The c-SM and c-PET/c-SM samples were observed to have similar thermal behavior under pyrolysis conditions. The onset decomposition temperature has been reported to indicate the thermal stability of a material under specified conditions. In the results in Figure 9a, it can be seen that the c-PET/d-coal samples showed a higher thermal stability under pyrolysis conditions, which may be due to a lesser amount of volatile compounds present in the c-PET/d-coal samples compared to their d-coal counterparts. In all samples, the peak decomposition temperature in argon was found to lie between 700 and 800 °C, which is attributed to the complete removal of volatile compounds such as C_x_H_y_, NO_x_, and SO_x_ via different pathways such as cracking and chain scission reactions [47]. The region of complete decomposition under pyrolysis conditions is typically followed by the formation of a char residue, which shows little to no changes in weight loss at high temperatures. In Figure 9a, it can be seen that all samples undergo negligible weight losses at temperatures above 800 °C, indicating the formation of a char residue up to 1000 °C. It can also be seen that the c-PET/c-ROM blend and c-PET/c-BCF had a higher residue yield in comparison to c-ROM blend and c-BCF, respectively. This may indicate a greater amount of fixed carbon obtained in the c-PET/d-coal samples compared to their d-coal counterparts of similar ash content.

In Figure 9b, the combustion behavior of each sample is presented, showing the thermal stabilities of the samples when exposed to air at increasing temperatures. It is important to note that the TG analysis was performed to resemble the ASTM D3174 standard, and hence a plot of mass% versus time is presented in Figure 9b. The standard method states that in the first 60 min., the temperature must reach 500 ± 10 °C and then reach 950 ± 20 °C in the next 60 min. A further 120 min. is required for holding the sample at 950 ± 20 °C to complete the test. From the results in Figure 9b, it can be seen that the onset decomposition temperatures (i.e., T at 5% weight loss) in air were all ca. 50 min., were around 500 °C. Under combustion conditions, the main reactions leading to the decomposition of the carbonaceous sample can be predominantly from incomplete combustion to form carbon monoxide (CO) and the rate of decomposition may depend on the type of hydrocarbons present in the sample. A large amount of aromatic compounds may result in faster combustion rates due to their high heat content in comparison to aliphatic hydrocarbons of the same carbon atoms [48]. The c-PET/d-coal samples were found to undergo complete decomposition around the 150 min. mark, whereas the c-ROM blend and c-SM underwent complete decomposition at 175 min. and 280 min., respectively. This showed a greater thermal stability at 950 °C and slow combustion kinetics in c-ROM blend and c-SM samples compared to their c-PET/d-coal counterparts. The c-PET/c-BCF and c-BCF samples resembled similar combustion behavior as they showed similar rates of the decomposition. The observed differences can be attributed to the amount of aromatic hydrocarbons present in each sample. It has been shown that aromatics such as naphthenes show greater combustion rates than their aliphatic counterparts with the same number of carbon atoms [48]. The carbonization of w-PET has also been shown to results in a char product predominantly consisting of aromatic compounds and some degree of graphitic carbon in its fixed carbon content [36]. The result in Figure 9b may therefore indicate an increased amount of aromatic compounds in c-PET/d-coal samples after co-carbonization compared to the d-coal samples.

The fixed carbon content of the samples was, therefore, calculated using the same principles for both sets of results, i.e., sample weight loss under inert conditions gives the volatile matter content of the sample whereas the sample residue remaining after complete combustion in air gives the ash content (also see Equation (S1)). A summary of the calculated ash content, volatile matter content, and fixed carbon content for carbonized samples is presented in Figure 10.

In Figure 10a–d, it can be seen that there is good agreement between the ash, fixed carbon, and volatile matter content values measured using both TGA and ASTM methods. In Figure 10c, the variations in volatile matter amongst some of the samples may be attributed to experimental error such as inaccurate moisture measurements, but for the purposes of this study, the moisture content was included as part of the volatile matter measurements in TGA experiments hence such errors could be expected. The results in Figure 10a show that the ash content trend for d-coal samples and their c-PET/d-coal counterparts is as follows: c-SM >> c-BFC > c-ROM blend. Using the data in Appendix A, it can be seen that there is a significant difference of about 36% between the ash content of c-PET/c-SM in comparison to c-SM, about 10% between c-PET/c-BFC and c-BFC, with only a 3% difference between c-PET/c-ROM blend and the c-ROM blend. The trend for the fixed carbon content, on the other hand, is as follows: c-ROM blend >> c-BFC > c-SM, with the same holding for the c-PET/d-coal composites. In Figure 10c, the volatile matter content remains relatively the same across the c-PET/d-coal samples and slightly differs between the carbonized d-coal samples, however, with clear indication that the c-PET/d-coal samples have a lower volatile matter content in comparison to their carbonized d-coal counterparts. Interestingly, in Figure 10d, the amount of additional fixed carbon content, i.e., carbon derived from carbonization of w-PET during the co-carbonization of a 3:1 w-PET/d-coal mixture, follows a trend similar to the ash content trend in Figure 10a. In other words, the sample with the highest ash content, c-SM, was found to yield the highest additional fixed carbon content upon its co-carbonization with w-PET. Since all w-PET/d-coal mixtures were co-carbonized in a 3:1 ratio (w-PET:d-coal), the results in Figure 10d clearly indicate that the amount of additional fixed carbon is proportional to the ash content of the d-coal sample. Additionally, it is clear from results obtained in Figure 4 that there is a 48% threshold for the amount of fixed carbon derived from w-PET when co-carbonized with d-coal ash under the process conditions specified in this study. In addition, it is also clear that the lower the ash content of the d-coal sample, the lesser will be the extent of additional fixed carbon, as evidenced for the c-PET/c-ROM blend and c-PET/c-BFC composites. The results, thus, clearly demonstrate the catalytic effect of d-coal ash in the conversion of w-PET to carbon under the conditions specified in this study.

All the c-PET/d-coal composites clearly show improved fixed carbon content compared to the c-ROM blend, c-BFC, and c-SM counterparts, but further to that, there were also visible differences in the physical appearance of the c-PET/d-coal composite in comparison to the both raw d-coal and carbonized d-coal samples.

### 3.3. Physical and Mechanical Properties of c-PET/d-Coal Composites

In Figure 11, the difference between the physical appearances of raw BFC are compared to heat-treated BFC, and the co-carbonized c-PET/c-BFC composite. It can be seen that the heat treatment of raw BFC at 700 °C under nitrogen flow did not change its physical appearance as it remained in powder form; however, the c-PET/c-BFC product was converted into compact agglomerates that retain the shape of the crucible boat. The bulk of the agglomerate structure was highly porous and grey in appearance, resembling the physical appearance of metallurgical coke [7,49,50]. The same result was true for the ROM blend, SM, and their c-PET/d-coal counterparts. This phenomenon can be attributed to an induced caking process during the co-carbonization of w-PET/d-coal composites, which either occurs to a very low degree or not at all in the raw d-coal samples.

The caking process of coking coals occurs upon heat treatment under inert conditions. The process is characterized by three main stages, namely, plastic deformation, softening, and resolidification, which are detailed in the literature [51,52]. The caking ability of w-PET/BFC can be explained by taking into account the thermal behavior of w-PET upon its exposure to elevated temperatures. The DSC analysis (Appendix A) showed that w-PET undergoes a glass transition around 120 °C followed by melting at 260 °C, consistent with the findings of Ko et al. [36]. At the glass transition temperature, w-PET chains soften and have increased fluidity due to chain mobility at temperatures above the glass transition. A greater degree of PET chain fluidity occurs as the w-PET viscosity further decreases approaching the melting temperature range. In addition, in-depth studies [29,53] on the decomposition of PET (around 450 °C) have reported on the formation of short-chain waxes as by-products, and these can also increase fluidity within the w-PET/d-coal composites. During the co-carbonization of w-PET/d-coal composite, the w-PET particles can be assumed to act as a modifier/plasticizer, similar to coal tars and oils [49,54], thereby increasing the coal fluidity resulting in improved caking ability of the d-coals. The solidification stage can be attributed to the formation of a solid carbon (c-PET) residue at 700 °C.

The observed caking ability of w-PET/d-coal composites was found to give the material structural integrity such that the compressive strength of the material could be measured. In Figure 12, the compressive strength of c-PET/c-ROM blend is compared to that of commercial metallurgical coke, according the ISO 4700:2007 standard testing method.

In Figure 12, it can be seen that the compressive strength of c-PET/c-ROM blend had a maximum load of up to 98% less compared to metallurgical coke. This is attributed to the highly porous structure of the c-PET/c-ROM blend composite (as seen in Figure 11b) which renders the c-PET/c-ROM blend composite particles very fragile/brittle. The highly porous nature may result from the release of gaseous by-products during co-carbonization at 700 °C and possibly the lack of compaction of w-PET/ROM blend samples during their co-carbonization. In contrast, the cross-sectional surface of metallurgical coke particles is more compact and shows very little porosity since the precursor coal particles are compacted together during a conventional coke-making process in a coke oven. The compressive strength results are important when considering reductants used in the steel making industry; however, for some ferroalloy operations the most important reductant properties are high fixed carbon content and low ash, which were achieved in this current study.

The mechanical properties become important in smelting operations where a coke bed is formed, to which a burden of mineral ore lumps are stacked on top of the coke bed [55]. In such operations, such as submerged arc furnaces and blast furnaces, the use of a coke material with poor compressive strength would cause the generation of fines which may lead to blockages in the extraction ducts and feed ports. In addition, the loss of coke particles through fines generation may lead to a lowered reducing potential and generate less alloy than desired. It is, however, shown that in open-arc operations, the need for a highly strong and mechanically robust coke is not of great importance since there is continuous formation of a molten bath [56]. There is little to no burden formed under open-arc operations and the most important property of coke becomes the fixed carbon content and to a lesser extent the mechanical strength. The results obtained in this study show that a higher fixed carbon content was achievable in c-PET/d-coal samples reaching as high as 85% fixed carbon, typically desired for open-arc smelting operations.

## 4. Conclusions

The aim of the study was to investigate the co-carbonization of w-PET with d-coal at 700 °C in an attempt to obtain a carbonized c-PET/d-coal composite with a higher fixed carbon content and lower ash content in comparison to the d-coals treated under the same carbonization conditions. The results obtained in this study show that d-coals derived from non-coking coals could be co-carbonized in a 3:1 w-PET:d-coal mass ratio to obtain a carbonized composite material (c-PET/d-coal) with improved fixed carbon content and lower ash and volatile matter content in comparison to d-coal alone. In addition, the c-PET/d-coal composite undergoes a caking process during co-carbonization such that the final product attains an agglomerated physical structure without the use of a binder. The ash and fixed carbon content of a c-PET/c-ROM blend composite were found to reach 7% and 86%, respectively, values that are typical for metallurgical coke. The results in this study also demonstrated the effect of d-coal ash on the extent of thermos-catalytic conversion of w-PET to solid carbon. It was found that the higher the ash content in a d-coal sample, the greater the amount of carbon formed from w-PET. The chemical composition of the carbon derived from co-carbonization of w-PET/d-coal composites was also found to be higher in degree of graphitization when compared to carbon obtained from direct carbonization of w-PET, where carbonization was carried out at 700 °C. The abundance of catalytic metal oxides in d-coal ash makes d-coals suitable to composite with w-PET and potentially develop an eco-friendly metallurgical coke alternative material in production of some ferroalloy open-arc operations that require high fixed carbon content and low ash content.

In addition to the results reported in this study, it is highly recommended that the co-carbonization of w-PET/d-coal be developed at pilot scale using conventional coke-making processes and the c-PET/d-coal product undergoes full characterization on its mechanical and reactivity properties such as the coke strength, coke reactivity index (CRI), and coke reactivity strength (CSR). Furthermore, recommendations by Zheng et al. [39] can be applied in the characterization of the c-PET/d-coal product in order to develop a coke molecular structural model based on techniques such as elemental analysis, Fourier transform infrared (FTIR) spectroscopy, Raman spectroscopy, XRD, high-resolution transmission electron microscopy (HRTEM), 13C nuclear magnetic resonance (NMR) imaging, and X-ray photoelectron spectroscopy (XPS). Application of such a coke molecular model can inform the prediction and optimization of the coke performance of the c-PET/d-coal composite and other non-coking or semi-coking coals.

## Figures and Tables

**Figure 1 materials-16-02782-f001:**
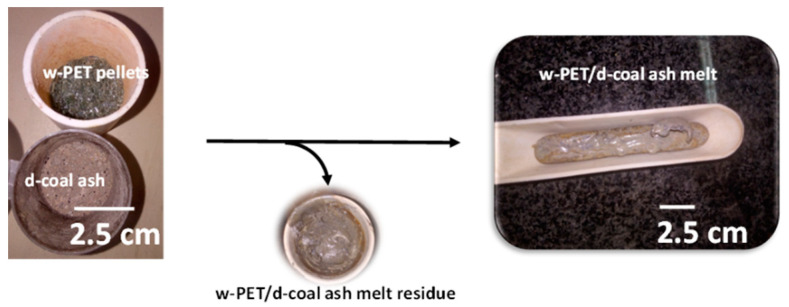
Photographs of w-PET pellets, d-coal ash powder, and w-PET/d-coal ash melt; also shown is the loss of material (melt residue) during transfer of the w-PET/d-coal ash melt to a ceramic boat.

**Figure 2 materials-16-02782-f002:**
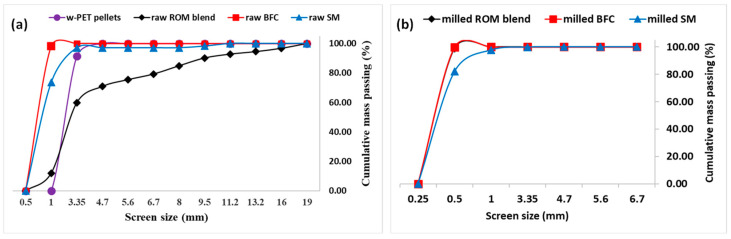
Particle size distribution (PSD) curves obtained from screened w-PET pellets and d-coal samples: (**a**) raw w-PET and raw d-coal samples and (**b**) milled d-coal samples. All the results are based on a dry basis analysis.

**Figure 3 materials-16-02782-f003:**
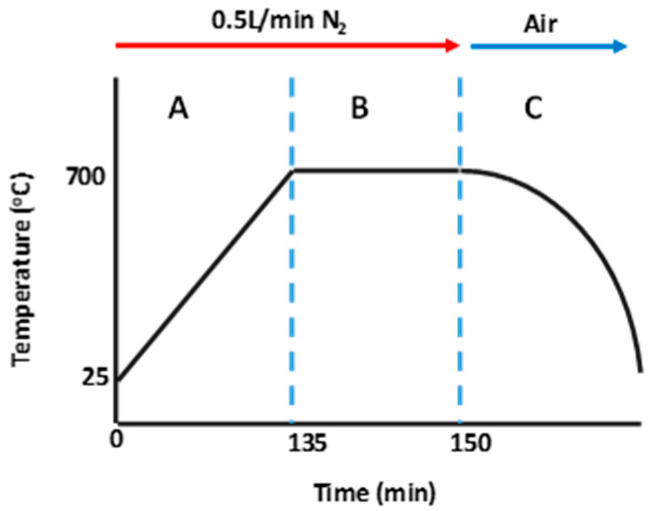
Schematic representation of the carbonization process used to prepare carbon derived from w-PET, w-PET/d-coal ash, and w-PET/d-coal samples.

**Figure 4 materials-16-02782-f004:**
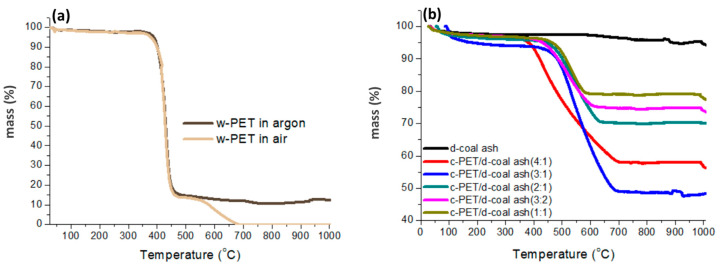
Thermal decomposition of w-PET and resultant c-PET from direct carbonization and co-carbonization with d-coal ash: (**a**) TGA thermogram of raw w-PET under an argon flow of 3 L/min (and static air) at heating rate of 10 °C/min up to 1000 °C and (**b**) TGA thermograms of different c-PET/d-coal ash composites from ambient temperature to 1000 °C under static air and a heating rate of 10 °C/min.

**Figure 5 materials-16-02782-f005:**
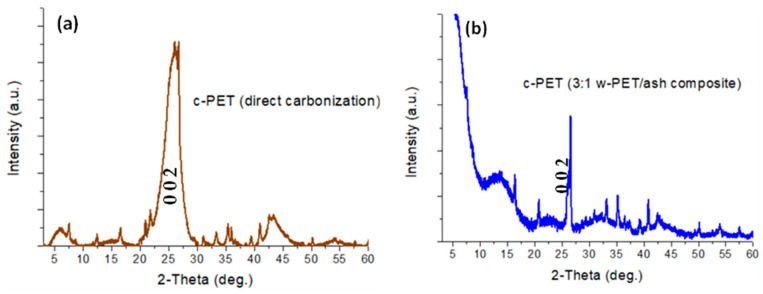
PXRD pattern of c-PET powder samples measured from 2-Theta 3–60° at a scan rate of 2°/min: (**a**) c-PET from direct carbonization of w-PET; (**b**) c-PET from co-carbonization of 3:1 w-PET/ash composite.

**Figure 6 materials-16-02782-f006:**
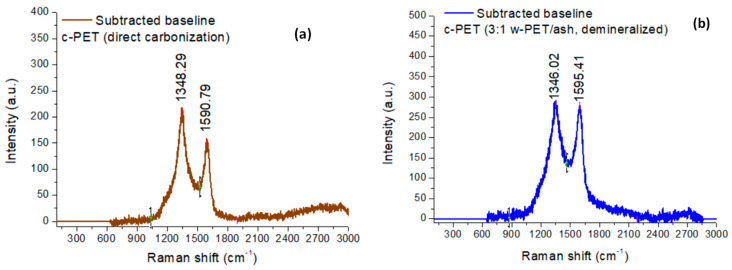
Raman spectra measured on c-PET powder samples from 100–3000 cm^−1^ at an excitation wavelength of 532 nm: (**a**) c-PET from direct carbonization of w-PET and (**b**) c-PET from co-carbonization of the 3:1 w-PET/d-coal ash composite. The selected peaks presented were baseline corrected/subtracted from the original spectra in Appendix A.

**Figure 7 materials-16-02782-f007:**
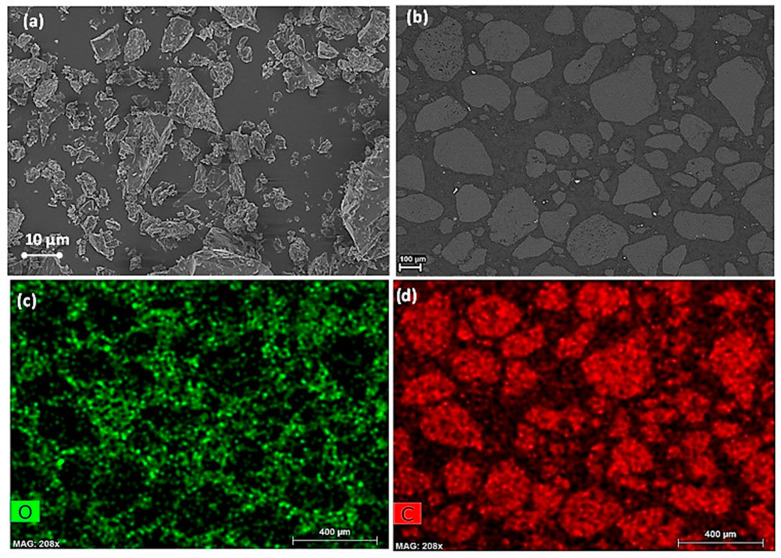
Microstructure analysis and elemental composition/distribution in c-PET particles obtained by direct carbonization of w-PET: (**a**) FIB-SEM image showing the 3D surface of c-PET particles (at 1000× magnification); (**b**) SEM image of the cross-sectional area of c-PET particles generated from image (**b**); (**c**) SEM-EDX elemental map of elemental oxygen atoms; and (**d**) SEM-EDX elemental map of elemental carbon atoms generated from image (**b**).

**Figure 8 materials-16-02782-f008:**
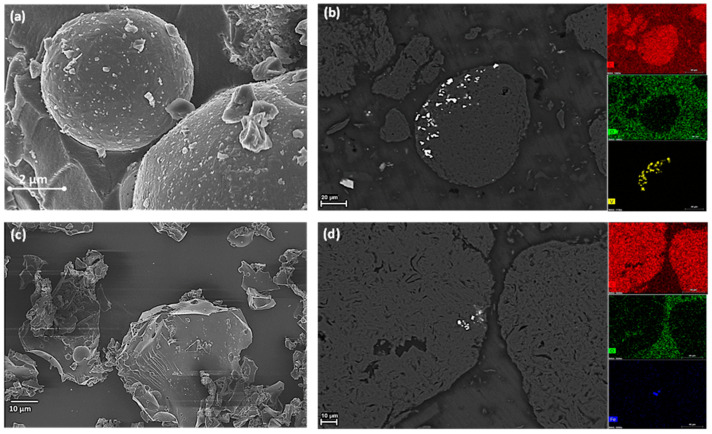
Morphology of de-mineralized c-PET/d-coal ash composite particles derived from co-carbonization of a 3:1 w-PET/d-coal ash composite: (**a**) FIB-SEM image showing 3D surface of spherical particles (at 10,000× magnification); (**b**) SEM image of the cross-sectional surface of spherical particles; (**c**) FIB-SEM image showing 3D image of irregularly shaped particles (at 1000× magnification); and (**d**) SEM images of the cross-sectional area of irregularly shaped particles.

**Figure 9 materials-16-02782-f009:**
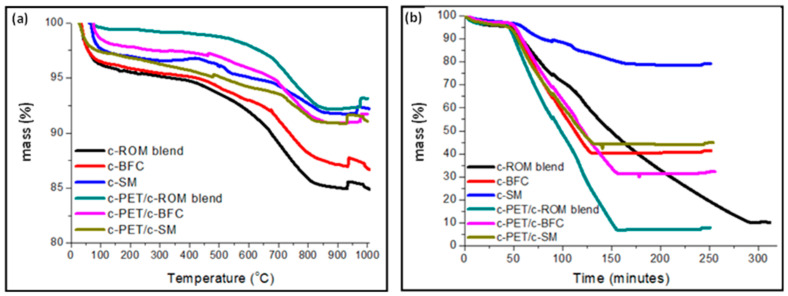
TGA thermograms for d-coal samples and their c-PET/d-coal composite counterparts. Each sample was carbonized at 700 °C under constant nitrogen flow prior to analysis: (**a**) thermal analysis under Argon (Ar) flow at a flowrate of 3 L/hour and heating of 10 °C/min up to 1000 °C and (**b**) thermal analysis in static air at 10 °C/min up to 1000 °C (reached after 100 min) and held at 1000 °C until no weight change was observed.

**Figure 10 materials-16-02782-f010:**
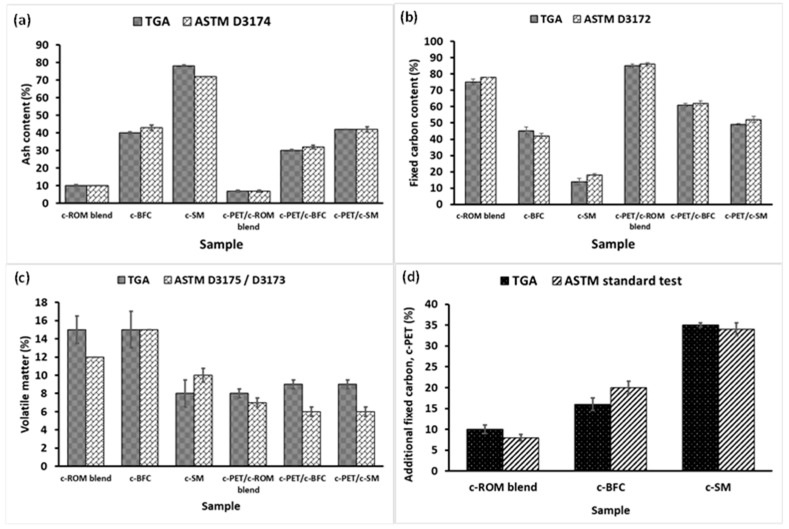
Graphical comparison of the fixed carbon content and additional c-PET obtained for each c-PET/d-coal composite after co-carbonization at 700 °C: (**a**) ash content as per ASTM D3174 and TGA/DSC analysis, (**b**) fixed carbon content as per ASTM D3172, (**c**) volatile matter as per ASTM D3175/D3173, and (**d**) additional PET-derived carbon, c-PET, calculated as the difference between the fixed carbon content of c-PET/d-coal and carbonized d-coal.

**Figure 11 materials-16-02782-f011:**
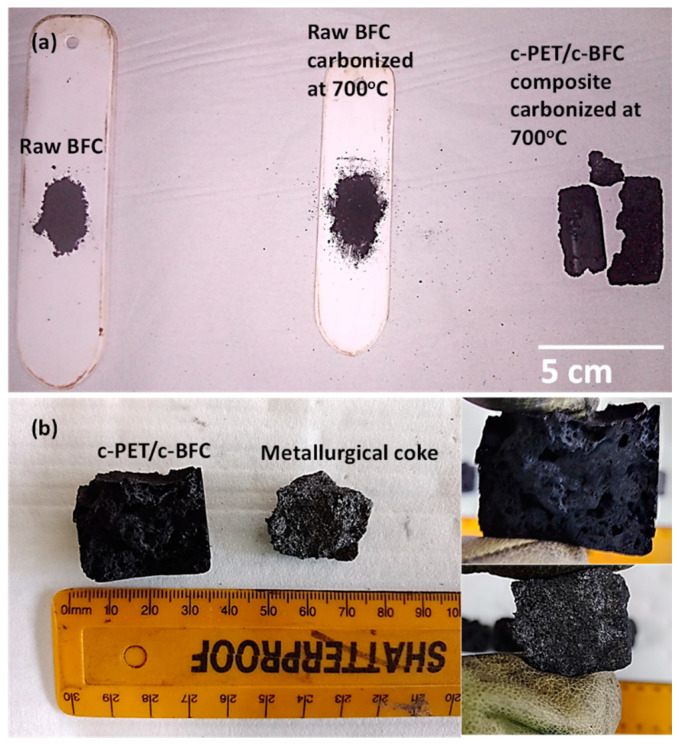
Physical appearance of raw BFC and carbonized BFC and c-PET/c-BFC composite: (**a**) comparison of non-carbonized and carbonized BFC with c-PET/c-BFC composite and (**b**) cross-sectional surface appearance of c-PET/c-BFC compared to metallurgical coke.

**Figure 12 materials-16-02782-f012:**
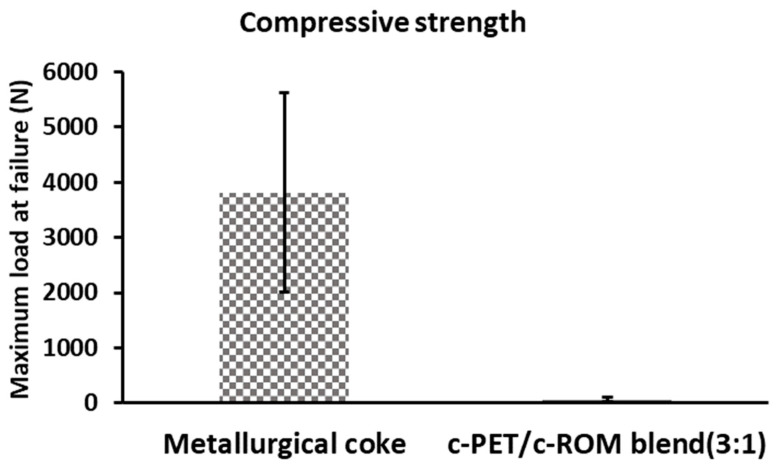
Bar graph comparing the compressive strength results obtained for commercial metallurgical coke and c-PET/c-ROM composite. The results are only based on the maximum load at failure.

**Table 1 materials-16-02782-t001:** Summary of the TGA results (residue yield, volatiles, and fixed carbon) obtained for co-carbonization products derived from different w-PET/d-coal ash composites.

	Composites					d-Coal Ash
Blend Ratios	w-PET/d-Coal Ash (4:1)	w-PET/d-Coal Ash (3:1)	w-PET/d-Coal Ash (3:2)	w-PET/d-Coal Ash (2:1)	w-PET/d-Coal Ash (1:1)	
Residue (*R*) (wt%)	56	48	70	77	74	94.2
Volatiles (*V*) (wt%)	44	52	30	23	26	5.8
[*C*]*_PET_* (wt%) ^#^	38.2	46.2	24.2	17.2	20.2	0

^#^ The c-PET yield is calculated using Equations (1) and (2).

## Data Availability

The data presented in this study are available in this article and Appendix A.

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
