# Peer review of "Co-Carbonization of Discard Coal with Waste Polyethylene Terephthalate towards the Preparation of Metallurgical Coke"

_materials, 2023, doi:10.3390/ma16072782_

Round 1
Reviewer 1 Report
This manuscript presents the authors’ efforts to reduce environmental pollution with polyethylene terephthalate wastes mixed with stockpiled discard coal from waste pits, by transforming it into carbon enriched composites with useful properties similar to the metallurgical coke. The method chosen by the authors of the article is that of co-carbonization of plastic in the presence of coal dust at 700oC, in an inert atmosphere (nitrogen). The subject is relevant to the field. It is addressed to the specialists concerned with the disposal of coal waste with high ash content resulting from mining operations. These wastes become dangerous for human health polluting the soil, and presented a long-term risk of spontaneous combustion in the storage pits. Although the idea of ​​using plastic waste with different chemical structures to increase the carbon content of materials is not very new, still improving the properties of coal waste by co-carbonization in the presence of discarded polyethylene terephthalate is an excellent idea that can be recommended for future research. Although the idea of using synthetic plastic waste with different chemical structures to increase the carbon content of materials is not very new, however, attempts to obtain useful materials from polluting waste by treatment at high temperatures is an excellent idea that can be developed in future research.
The literature cited in the manuscript reflects the state of research in the field.
Unfortunately, some experimental aspects are unclear and difficult to reproduce experimentally. I am referring in particular to the part that describes the thermal treatments of materials in order to obtain composites. My opinion is that respective sections should be completely rewritten to be well understood by the readers. Other aspects that should reviewed:
2.2.1 Section Direct combution of w-PET, row 158: The reating rate is an important parameter when discussing about the thermal processes. The heating rates between 5 and 10oC min-1 is too large to be included in a scientific paper. The authors should decide on a certain value of the heating rate. The values should be specified more precisely in the other sections of the manuscript.
2.2.2. Section, Co-carbonization of w-PET with discard coal ash rows 179-181: The uniform distribution of d-coal particles in w-PET matrix is only assumed by the authors and not proven.
2.2.3. Post-synthesis acid de-mineralization of c-PET/d-coal ash product. This title should be changed because it is about a purification process and not by a synthesis.
2.2.4 Section, Please specify the device with which the sample was ground (name of the producing company, country, city).
3.1. Section, Direct carbonization of w-PET in comparison to co-carbonization of w-PET/d-coal ash composites (page 7 ): The Y axis in the thermograms shown in figures 3 (a) and 3(b) must be mass or mass(%) and not Weight change (%). Please correct. The same correction must be applied to the thermograms in figures S-1 and S-6 and Figures 8a and 8b. The equipment with which TG investigation were made must be specified in the text along with name of company, producing country and the city.
The conclusions are clear and correctly argued instrumentally.
Reviewer 2 Report
I have read the article entitled “Co-carbonization of discard coal with waste polyethylene terephthalate towards the preparation of metallurgical coke” Changes need to be made to the manuscript before acceptance.
Line 133-139 The objective of the study is not clear in addition to the percentage by added weight in which it is different from the study of Nomura et al.
2.2.1 Add the equation with which the percentage of residue is calculated.
Line 341 Change Bands by Plane
Line 344 Change band by plane
Figure 4 Place the 002 plane on the graph
Figure 8 Discuss each weight loss, compare with what is reported add references.
Discuss how mechanical properties affect the material obtained compared to what is reported in the literature
Round 2
Reviewer 1 Report
The authors completed the manuscript according to the reviewer's recommendations.
Author Response
The authors accept the decision of the reviewer and would like to thank them for their time taken to review the manuscript.
Reviewer 2 Report
the authors made all the suggested changes therefore it is accepted in the present form
Author Response
The authors accept the decision of the reviewer and would like to thank them for the time taken to review the manuscript.